

# Diversity of bioprotective microbial organisms in Upper Region of Assam and its efficacy against *Meloidogyne graminicola*

Rupak Jena[1,2], Bhupendranath Choudhury[1], Debanand Das[1], Bhabesh Bhagawati[1], Pradip Kumar Borah[1], Seenichamy Rathinam Prabhukartikeyan[2], Swoyam Singh[3], Manaswini Mahapatra[4], Milan Kumar Lal[5], Rahul Kumar Tiwari[5] and Ravinder Kumar[5]

[1] Department of Nematology, Assam Agricultural University, Jorhat, Assam, India
[2] Division of Crop Protection, National Rice Research Institute, Cuttack, Odisha, India
[3] Department of Entomology, Siksha O Anusandhan, Bhubaneswar, Odisha, India
[4] Department of Agriculture and Allied Sciences (Plant Pathology), C.V. Raman Global University, Bhubaneswar, Odisha, India
[5] Division of Plant Protection, ICAR-Central Potato Research Institute, Shimla, Himachal Pradesh, India

Corresponding authors
Rupak Jena, rupu.jena27@gmail.com
Ravinder Kumar, chauhanravinder97@gmail.com

## ABSTRACT

*Meloidogyne graminicola* has a well-established negative impact on rice yield in transplanted and direct-seeded rice, resulting in yield losses of up to 20 to 90 percent. Studies were undertaken to isolate potential native strains of bio-control agents to manage the devastating Rice Root Knot Nematode (*M. graminicola*). Eighteen bacterial strains and eleven fungal strains were isolated from the rhizosphere of crops like rice, okra, ash gourd, chili, beans and cucumber, enveloping diverse soil types from the Upper Brahmaputra Valley region of Assam. Six bacterial strains were gram-positive according to morphological results, while twelve others stained negatively. Fifteen bacteria were rod-shaped, two were coccus and one was diplococcus, and all the bacterial isolates showed signs of movement. All the bacterial strains exhibited positivity for gelatin hydrolysis and catalase test. Seven bacteria showed positive, while eleven showed negative reactions to possess the ability to deduce carbon and energy from citrate. The study of the *in vitro* efficacy of the twenty-nine bacterial and fungal isolates tested against second-stage juveniles ($J_2$) of *Meloidogyne graminicola* revealed that all the bacterial and fungal isolates potentially inhibited the test organism and caused significant mortality over sterile water treatment. The promising bacterial and fungal isolates that exhibited mortality above 50% were identified as BSH8, BTS4, BTS5, BJA15, FJB 11 and FSH5. The strain BSH8 exhibited the best result of mortality, with 80.79% mortality against $J_2$ of *M. graminicola*. The strain BTS4 and BTS5 expressed mortality of 71.29% and 68.75% under *in-vitro* conditions and were significant. The effective and promising bioagents were identified using the 16 S rRNA sequencing as *Bacillis subtilis* (BSH8), *Bacillus velezensis* (BTS4), *Alcaligenes faecalis* (BTS5), *Rhizobium pusense* (BJA15), *Talaromyces allahabadensis* (FSH5) and *Trichoderma asperellum* (FJB11). These results indicated the microorganism's potential against *M. graminicola* and its

potential for successful biological implementation. Further, the native strains could be tested against various nematode pests of rice in field conditions. Its compatibility with various pesticides and the implication of the potential strains in integrated pest management can be assessed.

# INTRODUCTION

Rice (*Oryza sativa*) is a crucial food crop that plays a significant role in the world's socioeconomic scenario. It nourishes around four billion people, spreading over 160 million hectares. The significance of this crop can be measured by the fact that it contributes to 20% of the total calorie intake globally (*Le, 2010*). Rice is widely recognized as an important dietary cereal crop, ranking second in its growing area (*Devi & Ponnarasi, 2009*). The crops deliver approximately 67% and 34% of the energy input for nearly three billion people in eastern Asia and 1,500 million in Latin America and Africa, respectively. In India, rice covers a major share of the food plate for about 800 million people, fulfilling the 43% calorie requirement and contributing nearly 40% and 55% to the country's food grain and cereal production, respectively (*Pathak et al., 2020*). India ranks 9th in productivity ratio, with the average yield across India estimated to be approximately 2.7 thousand kg/ha (https://www.statista.com/statistics/764299/india-yield-of-rice/). Lagging productivity in rice cultivation can be attributed to various biotic and abiotic factors. Biotic factors, such as the resurgence of insects and pests, including nematodes, threaten rice crops. Plant parasitic nematodes alone result in a staggering 21.3% crop loss, equivalent to approximately 102,039.79 million rupees annually (*Kumar et al., 2020*). Over 210 species of plant parasitic nematodes have been reported to be associated with rice crops (*Prot, Soriano & Matias, 1994*). The economically important nematodes associated with the crop are *Meloidogyne graminicola, Dictylenchus angustus, Heterodera oryzicola, Aphelenchoides besseyi, Hirschmanniella* spp. *Dictylenchus angustus* and *Aphelenchoides besseyi* feed on the aboveground parts. Rice root-knot nematode (*M. graminicola*), stem nematode (*Ditylenchus angustus*), white tip nematode (*Aphelenchoides besseyi*), and cyst nematode (*Heterodera oryzicola*) were collectively projected to cause yield losses of 10.50 percent and losses of 779.30 million rupees, respectively (*Jain, Mathur & Singh, 2007*).

*M. graminicola* is a sedentary obligate endo-parasitic nematode, and its female reproduces by meiotic parthenogenesis and amphimixis. It lays eggs (200–500) in a gelatinous matrix inside the root. First-stage juveniles (J1) are produced after 4–7 days, and they molt into second-stage juveniles in the next 2–3 days ($J_2$). Under favorable environmental conditions, the infectious $J_2$ is poised to hatch out of the egg. The $J_2$ is the infective stage, and it enters through the elongation zone and induces the formation of syncytium and gall (*Abad et al., 2003*). The loss caused by *M. graminicola* alone was reported at 16–32% in irrigated and 11–74% in flooded and submerged paddy in India

(*Soriano, Prot & Matias, 2000*). Overall, *M. graminicola* has a well-established negative impact on rice yield, resulting in yield losses of up to 20 to 90 percent. The pathogenic nematodes in soil act as primary invaders against agricultural crops and supplement the secondary pathogens like bacteria, fungi, viruses, *etc.*, in establishing disease quadrangle. Using chemical pesticides has been a quick solution for managing nematodes. Still, the recent focus on sustainability and the harmful effects of pesticides has led to the need for a safer alternative. The shift towards more sustainable practices has made finding a safer solution for managing pests imperative.

Bio-control agents (BCA) are the potentially efficient microorganisms in the rhizosphere that can be harnessed and successfully implicated against harmful microorganisms (*Raymaekers et al., 2020*). Studies reported the successful application and identification of bioagents for the control of nematode plant diseases (*Nagendran et al., 2013*), however recently, efficient bio-control agents like *Pasteuria penetrans*, *Bacillus* sp., *Verticillium chlamydosporium*, *Paecilomyces lilacinus*, *Trichoderma* spp. *etc.* have shown promising results against various plant parasitic nematodes like *Meloidogyne* spp., *Aphelenchus avenae, Globodera* spp., *Pratylenchus* spp., *Heterodera* spp., *Rotylenchulus* spp (*TóthnéBogdányi et al., 2021*). Assam is an important state in India under the BGRIE (Bringing Green Revolution to Eastern India), with a total area of 2.28 million hectares under rice with a production rate of 4.86 million tons (*Bodh et al., 2019*) and parallel in fronting the menace of *M. graminicola* which threats the crop sustainability. Therefore, a study was undertaken to isolate and identify the promising native bioprotective rhizospheric microorganisms from the Upper Brahmaputra Valley region of Assam (UBVA) and, in succession, the *in vitro* efficacy of the isolated strains against the disease-causing second-stage juveniles of rice root-knot nematode *M. graminicola*.

## MATERIAL AND METHODOLOGY

### Maintenance of inoculum (*Meloidogyne graminicola*)

The pure culture of the rice root-knot nematode *Meloidogyne graminicola* was maintained in the susceptible rice variety Luit in pots in the Department of Nematology, Assam Agricultural University, Jorhat, India for its requisite during *in-vitro* evaluation. The seeds were obtained from ICR Farm, Assam Agricultural University, Jorhat, India; seedlings were germinated and then transplanted into 5 kg pots for inoculum maintenance.

### Collection of soil samples

A survey was conducted in the UBVA region, covering areas such as Alengmora, Sibsagar, Titabor, Jorhat, and Golaghat, in August 2021. The survey aimed to collect soil samples from the rhizosphere of different crops, including ash gourd, chili, okra, rice, beans, and cucumber (as listed in Table 1). Approximately 110 soil samples, weighing 100 grams each, were gathered during the survey. To ensure proper identification and analysis, the soil samples were carefully labeled with information such as the collection date, crop age, soil type, and crop variety. Subsequently, the samples were refrigerated at 5–10 °C until further examination.

**Table 1 Documentation on collection of rhizospheric soil samples from various crops in upper areas of Assam.**

| Sl. no | Code | Name of place | Village name | Code | Name of place | Village name |
|--------|------|---------------|--------------|------|---------------|--------------|
| 1. | BAK1 | Alengmora | Kahargaon | FAK1 | Alengmora | Kahargaon |
| 2. | BAK2 | Alengmora | Kahargaon | FTS2 | Titabor | ShyamGaon |
| 3. | BAB3 | Alengmora | Bahphola | FTS3 | Titabor | ShyamGaon |
| 4. | BTS4 | Titabor | ShyamGaon | FTK4 | Titabor | KarsoliGaon |
| 5. | BTS5 | Titabor | ShyamGaon | FSH5 | Sibsagar | HulalKalita |
| 6. | BTS6 | Titabor | ShyamGaon | FSS6 | Sibsagar | Salaguri |
| 7. | BTM7 | Titabor | Madhavpur | FSS7 | Sibsagar | Salaguri |
| 8. | BSH8 | Sibsagar | HulalKalita | FGM8 | Golaghat | Merapani |
| 9. | BSL9 | Sibsagar | LahonGaon | FJA9 | Jorhat | AAU |
| 10. | BSL10 | Sibsagar | LahonGaon | FJA10 | Jorhat | AAU |
| 11. | BGA11 | Golaghat | Amguri | FJB11 | Jorhat | Barbheta |
| 12. | BGA12 | Golaghat | Amguri | | | |
| 13. | BJA13 | Jorhat | AAU | | | |
| 14. | BJA14 | Jorhat | AAU | | | |
| 15. | BJA15 | Jorhat | AAU | | | |
| 16. | BJR16 | Jorhat | Rowriah | | | |
| 17. | BJR17 | Jorhat | Rowriah | | | |
| 18. | BJB18 | Jorhat | Barbheta | | | |

## Isolation of bacterial and fungal isolates

The soil samples collected from the crop's rhizosphere were thoroughly mixed individually and about 1 gm of the soil was weighed and taken for isolation of bacterial bioagents. The isolation of bacterial bioagents from soil involved serial dilution procedures. Sterile dilution blanks were marked sequentially, starting from stock and $10^{-1}$ to $10^{-6}$. A fresh sterile pipette transferred one ml from the stock to the $10^{-1}$ dilution blank. From a dilution tube of $10^{-7}$, 0.1 ml of dilution fluid was transferred into nutrient agar culture media. To facilitate the growth of bacteria, the plates containing soil fluid were sealed using paraffin wax and placed in an incubator set at a temperature of 28 ± 2 °C for 24 h. This incubation period allowed for the maximum recovery of bacterial colonies. After the 24-h incubation, distinct bacterial colonies with different morphologies were carefully chosen and subjected to repeated streaking. This process aimed to isolate pure bacterial colonies by ensuring the removal of any contaminants or mixed cultures (Table 1). A similar procedure was followed for the isolation of fungal bioagents. However, Potato Dextrose Agar (PDA) was employed as the culture medium instead of using nutrient agar. The plates containing PDA and soil samples were incubated for 3–4 days to support optimal fungal growth. After the incubation period, morphologically distinct fungal colonies were identified and selected. These selected colonies were then subjected to repeated streaking, enabling the isolation of pure fungal colonies (Table 1).

## Morphological and biochemical characterization of bacterial isolates

Colony characters like size, shape, the color of the colonies, and gram stain were recorded as cultural and morphological characters of the bacteria. A few biochemical tests such as KOH solubility test, starch hydrolysis test, citrate test, catalase test, and gelatin hydrolysis tests were also conducted for identification of the bacteria following protocols given in the Bergey's Manual of Determinative Bacteriology (*De Ley & Frateur, 1974*) (Fig. S1).

## Preparation of culture filtrates

Pure bacterial isolates were seeded to 100 ml of nutrient broth media and were incubated at 28 °C for 48 h. The liquid suspension was passed through Whatman No. 1 filter paper and once through a bacterial filter (*Aalten & Gowen, 1998*) and then centrifuged for 15 min at 6,000 rpm. The suspended residues were discarded once the supernatant was collected for testing. The same procedure was repeated for fungal bioagents, but potato dextrose broth was used as a seeding agent for pure fungal isolates instead of nutrient broth.

## *In vitro* efficacy of bacterial and fungal bio-agents

The extracted cell-free culture filtrates of isolated bacterial and fungal bio-agents were considered as a stock solution (100% concentration). The sterile water was added to the stock solution to have S/2, S/4, and S/8 concentrations of cell-free bacterial and fungal culture filtrates. About 2 mL of the cell-free stock culture filtrate obtained from the isolated bacterial and fungal bio-agents were carefully poured into sterile cavity blocks. Each cavity block was then supplemented with 50 newly hatched juvenile (J2) specimens of *M. graminicola*, which had been isolated overnight. A completely randomized block design (three factorial CRD) was employed to ensure a robust experimental design with three replications. All the cavity blocks were stored in the laboratory at room temperature. Observations on juvenile mortality were recorded at specific time intervals, namely 6, 12, 24, and 48 h after exposure to the bio-agent culture filtrates. The experimental treatments included various concentrations of the bio-agent culture filtrates, while sterile distilled water (SDW) was maintained as the control. To calculate the percentage of juvenile mortality, the following formula was utilized:

$$Percent\ Mortality = \frac{Number\ of\ dead\ juveniles\ in\ the\ treatment}{Total\ number\ of\ Juveniles\ in\ the\ treatment} \times 100$$

## Identification of efficient bio-agents

Bacterial and fungal isolates that displayed a minimum of 50% mortality in the *in vitro* assay were chosen for further analysis. Genomic DNA was extracted from these isolates using the EXpure Microbial DNA extraction kit from Bogar BioBee stores Pvt. Ltd (Coimbatore, India). The amplification of the genomic DNA was carried out using 16S rRNA-based primers. The forward primer used was 27F (5′ AGAGTTTGATCTGGCTCAG 3′), and the reverse primer was 1492R (5′ TACGGTACCTTGTTACGACTT 3′). The PCR products obtained from the bacterial isolates were sent to Triyat Scientific Co. Pvt. Ltd., (Maharashtra, India) for sequencing. Phylogenetic analysis was performed by comparing
**Table 2 Data on potential bacterial and fungal strains isolated from the crop rhizosphere and the crop characteristics.**

| Code | Name of place | Village name | Crop | Variety | Type of soil | Stage of the crop |
|------|---------------|--------------|------|---------|--------------|-------------------|
| BTS4 | Titabor | Shyam Gaon | Rice | Ranjit | Sandy loam (Submerged) | Flowering |
| BTS5 | Titabor | Shyam Gaon | Rice | Ranjit | Sandy loam (Submerged) | Flowering |
| BSH8 | Sibsagar | HulalKalita | Okra | Kranti | Loamy | Fruiting |
| BJA15 | Jorhat | Jorhat | Rice | Luit | Sandy Loam | Flowering |
| FSH5 | Sibsagar | HulalKalita | Okra | Kranti | Loamy | Fruiting |
| FJB11 | Jorhat | Barbheta | Chili | Baljuri | Loamy | Fruiting |

the obtained sequences with closely related sequences using MUSCLE 3.7 for multiple alignments (*Edgar, 2004*). The program Tree Dyn 198.3 was used to construct the phylogenetic trees based on the alignment results (*Dereeper et al., 2008*) (Table 2).

## Statistical analysis

The *in-vitro* three factorial completely randomized design statistical programs were used to analyze data on percent mortality statistically. The main effects' isolates, concentrations, times of exposure, and their interactions were assessed for important differences at $P < 0.05$. The percentage values were subjected to arcsin transformation and data were analyzed using SAS and IBM SPSS statistics 20.0 software. Tukey test and DMRT test were conducted to determine the significance of treatments. The experimental data obtained were analyzed using Fisher's Analysis of Variance method. The standard error of deviation (S.Ed) between the mean of the treatment combination was calculated as:

$$\text{S.Ed} \pm \ = \sqrt{\frac{2 \times \text{Error Mean Square}}{\text{Number of replication}}}$$

# RESULTS

## Isolation and characterization of bacterial bioagents from crop rhizosphere

Among the 110 soil samples from the Upper Brahmaputra Valley region of Assam, $n = 18$ potential spore-forming bacterial microorganisms and $n = 11$ potential fungal isolates were isolated. Based on colonial morphology, microscopy, and biochemical reactions, pure cultures were maintained. The spore-forming bacterial strains were selected for further confirmatory tests. The locality and village from where they were isolated, as well as their classification as bacteria (B) and fungus (F), were all taken into account while coding the bacterial and fungal isolates (Table 1).

**Table 3 Morphological and biochemical characteristics of the isolated strains of bacteria.**

| Isolated strain | Colour | Gram staining | Shape | Motility | KOH | Citrate utilization | Gelation hydrolysis | Catalase test |
|---|---|---|---|---|---|---|---|---|
| BAK1 | White | +ve | Rod | + | −ve | −ve | +ve | +ve |
| BAK2 | Creamish yellow | −ve | Coccus | + | +ve | +ve | +ve | +ve |
| BAB3 | Creamish yellow | −ve | Rod | + | +ve | +ve | +ve | +ve |
| BTS4 | White | +ve | Rod | + | −ve | +ve | +ve | +ve |
| BTS5 | Yellow | −ve | Rod | + | +ve | −ve | +ve | +ve |
| BTS6 | Whitish | −ve | Coccus | + | +ve | +ve | +ve | +ve |
| BTM7 | Whitish | −ve | Rod | + | +ve | +ve | +ve | +ve |
| BSH8 | Yellow | +ve | Rod | + | −ve | +ve | +ve | +ve |
| BSL9 | Yellow | +ve | Rod | + | −ve | −ve | +ve | +ve |
| BSL10 | White | +ve | Rod | + | −ve | −ve | +ve | +ve |
| BGA11 | White | −ve | Rod | + | +ve | +ve | +ve | +ve |
| BGA12 | Yellow | −ve | Rod | + | +ve | +ve | +ve | +ve |
| BJA13 | White | −ve | Rod | + | +ve | +ve | +ve | +ve |
| BJA14 | White | −ve | Rod | + | +ve | +ve | +ve | +ve |
| BJA15 | White | −ve | Rod | + | +ve | −ve | +ve | +ve |
| BJR16 | White | +ve | Diplococcus | + | −ve | +ve | +ve | +ve |
| BJR17 | Yellow | −ve | Rod | + | +ve | −ve | +ve | +ve |
| BJB18 | Creamy white | −ve | Rod | + | +ve | −ve | +ve | −ve |

## Morphological and biochemical characterizations of isolated bacterial strains

### Morphological characterization

Terminal and subterminal spores were segments of the 18 motile bacterial isolates, of which 15 were rod-shaped, two were coccus-shaped, and one was diplococcus-shaped. The isolated bacteria exhibited diverse colony characteristics resembling regular, irregular, slightly raised, flat, and colors, such as whitish, creamish white, creamish yellow, yellowish orange, and others. Bacterial isolates BAK2, BAB3, BTS5, BTS6, BTM7, BGA11, BGA12, BJA13, BJA14, BJA15, BJR17, and BJB 18 were gram-negative in response to the gram staining test, while strains BAK1, BTS4, BSH8, BSL9, BSL10, and BJR16 tested positively (Table 3).

### Biochemical characterization

KOH supplements and replicates the gram staining results in which the 3% potassium hydroxide dissolves the cell wall of the gram-negative bacterial isolates leading to the string formation by the released viscous chromosomal material of the strain. The isolates BAK1, BTS 4, BSH8, BSL9, BSL10, and BJR 16 thick peptidoglycan deposition in the cell wall resist dismantling the cell wall, thus confirming its gram-positive identity.

The establishment of potential biocontrol strains characterizes by their ability to withstand challenging abiotic conditions. The citrate test demarks the strains with the potential to channel citrate into its carbon and energy source. The enzyme citrate permease

(citrase) facilitates the citrate into bacteria. Distinctive color variation of the media from Prussian blue to green signifies a positive response to the citrate test. All the bacterial isolates exhibited positivity in the gelatinase enzyme's production that hydrolyzes gelatin to amino acids *via* polypeptides. Amino acids are an essential component required in regulating the biochemical activities of the potential strains. The presence of gelatinase is marked by the medium transformation from semi-solid to liquid. The bacterial carbohydrate metabolism accompanies hydrogen peroxide ($H_2O_2$) as the self-destructive end product. The presence of catalase alleviates the impact by converting ($H_2O_2$) to water and oxygen. In the present study, all the bacterial isolates showed the presence of the catalase enzyme by producing bubbles (Table 3) (Fig. S1).

### *In vitro* bioassay of bacterial and fungal isolates against rice root-knot nematode

All the bacterial cultural filtrates exhibited a marked and significant degree of larvicidal activity against the nematode compared to the control. Four bacterial isolates BSH8, BTS4, BTS5 and BJA15 exhibited mortality greater than 50%, but the culture filtrate of BSH8 (isolated from the rhizosphere of okra crop) showed average mortality of 80.79% against the infective juveniles of *M. graminicola*. After 48 h of exposure, the strain BSH8 triggered 100%, 96.67%, 95.33%, and 87.33% death at descending concentrations. The isolate BTS4, with an average mortality of 71.29% expressed a mortality of 97.33% at S concentration and 91.33% at S/2 concentration after 2 days of scrimmage. The culture filtrate of the strain BTS5 and BJA15 exhibited average mortality of 68.75% and 68.33% overall after 2 days of treatment against *Meloidogyne graminicola* (Graph1). The BSH8, BTS4, BTS5, and BJA15 strains were at par for the destruction of juveniles. The strain BGA11 isolated from rice showed the lowest potential for causing mortality, with 11.69% overall and 24% at S concentration after 48 h of exposure. Neglecting the cultural strain and exposure time, the mortality rate of the juveniles of *M. graminicola* at the concentration S/8 was 28.93%. In contrast, at concentration S, the mortality was exhibited to be a maximum of 52.78%. Similarly, irrespective of the cultural strain and concentration of the culture filtrate, the mortality was highest at 48 h after exposure at 50.61% and the minimal mortality was 25.96% 6 h after exposure time (Table 4 and Fig. 1).

The effectiveness of the isolated fungal strains was tested against *M. graminicola*, it became apparent that the fungal microorganisms substantially displayed mortality regardless of the different cultural filtrate concentrations and exposure times compared to controls (sterile water). Following *in-vitro* tests, FJB11 and FSH5 were the two most promising and efficient fungal strains and statistically significant. The fungal strain FJB11 isolated from the rhizosphere of chili exhibited a mortality mean of 78.83% against the juveniles of *M. graminicola* at all tested concentrations (Fig. 1). The strain FJB11 demonstrated alleviation of nematode mortality in various concentrations (S/8, S/4, S/2, and S) after 48 h of exposure. The results manifested 77.33%, 82.67%, 90.67%, and 100% mortality, with ascending concentrations, respectively. The strain FSH5, isolated from the rice rhizosphere, indicated promising results with an average nematode snuffing of 66.25%. The highest mortality was observed at the S concentration (90%) and S/2 concentration

**Table 4 Effect of culture filtrate of the isolated promising bacterial and fungal strains on the juvenile mortality of RRKN (*Meloidogyne graminicola*).**

| Treatments | Culture filtrate concentrate | Period of exposure | | | | Treatment (T) |
|---|---|---|---|---|---|---|
| | | 6 h | 12 h | 24 h | 48 h | |
| BTS4 (*Bacillus velezensis*) | S/8 | 39.33 (38.83) | 53.33 (46.92) | 64.00 (53.17) | 78.67 (62.51) | 71.29 (59.71) |
| | S/4 | 46.00 (42.70) | 58.00 (49.61) | 69.33 (56.41) | 83.33 (66.03) | |
| | S/2 | 60.67 (51.17) | 72.67 (58.50) | 85.33 (67.56) | 91.33 (73.26) | |
| | S | 72.67 (58.50) | 82.00 (64.92) | 86.67 (68.63) | 97.33 (80.73) | |
| BTS5 (*Alcaligenes faecalis*) | S/8 | 36.67 (37.27) | 47.33 (43.48) | 62.00 (51.95) | 78.67 (62.50) | 68.75 (58.79) |
| | S/4 | 41.33 (40.00) | 55.33 (48.07) | 70.00 (56.80) | 84.67 (66.96) | |
| | S/2 | 48.67 (44.24) | 62.00 (51.94) | 94.00 (75.95) | 99.33 (87.29) | |
| | S | 50.67 (45.38) | 72.00 (58.06) | 97.33 (80.73) | 100 (90.00) | |
| BSH8 (*Bacillus subtilis*) | S/8 | 51.33 (45.76) | 63.33 (52.73) | 75.33 (60.23) | 87.33 (69.17) | 80.79 (66.12) |
| | S/4 | 2.00 (51.95) | 70.67 (57.21) | 81.33 (64.40) | 95.33 (77.58) | |
| | S/2 | 76.67 (59.78) | 83.33 (65.91) | 92.00 (73.65) | 96.67 (79.60) | |
| | S | 75.33 (60.22) | 84.00 (66.44) | 98.00 (83.44) | 100 (90.00) | |
| BJA15 (*Rhizobium pusense*) | S/8 | 36.67 (37.26) | 49.33 (44.62) | 64.00 (53.13) | 75.33 (60.22) | 68.33 (58.07) |
| | S/4 | 42.00 (40.40) | 53.33 (46.91) | 66.67 (54.73) | 80.00 (63.45) | |
| | S/2 | 50.67 (45.39) | 60.00 (50.78) | 84.00 (66.52) | 99.33 (87.29) | |
| | S | 62.67 (52.34) | 76.00 (60.68) | 93.33 (75.28) | 100.00 (90.00) | |
| FSH5 (*Talaromyces allahabadensis*) | S/8 | 39.33 (38.83) | 48 (43.85) | 54.67 (47.68) | 64 (53.13) | 66.24 (54.97) |
| | S/4 | 50.67 (45.38) | 59.33 (50.42) | 65.33 (53.99) | 72 (58.15) | |
| | S/2 | 60 (50.79) | 67.33 (55.17) | 74 (59.41) | 82 (64.98) | |
| | S | 72 (58.09) | 77.33 (61.60) | 84 (66.53) | 90 (71.62) | |
| FJA11 (*Trichoderma asperellum*) | S/8 | 53.33 (46.91) | 62.67 (52.34) | 72 (58.06) | 77.33 (61.57) | 78.83 (64.18) |
| | S/4 | 63.33 (52.74) | 74.67 (59.81) | 77.33 (61.60) | 82.67 (65.45) | |
| | S/2 | 72.67 (58.52) | 79.33 (63.00) | 85.33 (67.55) | 90.67 (72.37) | |
| | S | 84 (66.53) | 89.33 (70.95) | 96.67 (79.60) | 100 (90) | |

**Note:**
Figures in parenthesis are arc signed values.

(82%) after 48 h of exposure. On average, the concentration effect showed the highest mortality at the S concentration (52.37%) and the lowest at the S/8 concentration (34.29%). The average effect of exposure time for the fungal isolates was highest at 48 h (50.73%) and lowest at 6 h (32.07%) (Table 4).

## Sequencing and phylogenetic analysis of bacterial and fungal isolates

The fingerprints of bacterial and fungal communities were generated by separating 16S rRNA gene fragments. The amplified PCR fragments of the efficient bacterial and fungal isolates containing 16S rRNA were subjected to analysis for homology using the n BLAST. The bacterial isolate BSH8 (OQ216891) was identified as *Bacillus subtilis* and was isolated from the okra rhizosphere (Fig. S2). The strains BTS4 (OQ216889), BTS5 (OQ216890), and BJA15 (OQ216892) were identified as *Bacillus velezensis*, *Alcaligenes faecalis*, and *Rhizobium pusense*, respectively, and were isolated from the rice rhizosphere (Fig. 2)

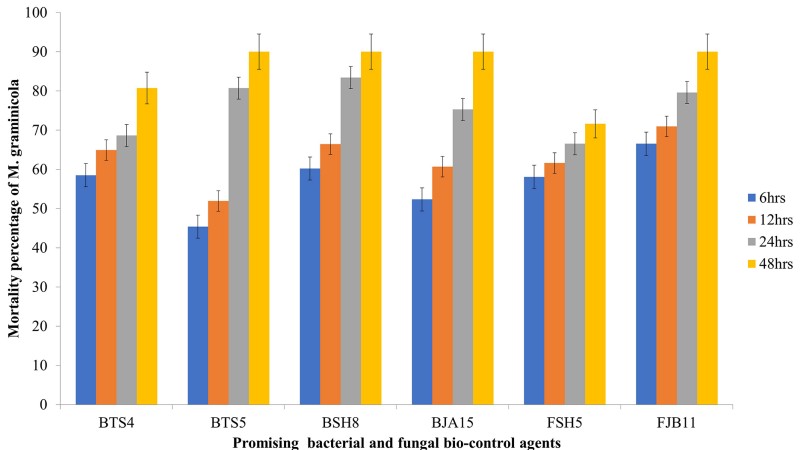

**Figure 1** Mortality rate of *M. graminicola* against the potential BCA after 6, 12, 24, 48 h in *in vitro* condition with 5% standard error (*P* < 0.05).

(Figs. S3–S5). Similarly, the fungal strains were also subjected to 16S rRNA identification. The bioagents FSH5 and FJB11 were reported to be *Talaromyces allahabadensis* (OQ244365)and *Trichoderma asperellum* (OQ244366) that were isolated from okra and chilli rhizosphere (Figs. S6 and S7). The isolate FSH5 also overlapped with the sequence of accession OM372948.1 and FJB11 was found to be 100% identical to the accession MN872484.1 (Fig. 3).

# DISCUSSION

Prevention of nematode disease progression requires feasible approaches supportable to the environment. Varying indiscriminate chemical applications in both quality and quantity to manage nematodes impairs soil health. Promising native microbial strains are becoming increasingly significant in nematode and pest management. The existence of countless and diverse advantageous microbial species in the earth's rhizosphere is enormous, and the studies undertaken are simply the tip of the iceberg (*Ciancio, Pietersen & Mercado-Blanco, 2019*). The plant holobiome hibernates majority of microbes and we extricated culturable 18 bacterial and 11 fungal promising microbes from 110 rhizospheric soil samples of agronomic crops such as rice, okra, cucumber, beans, ash gourd, and chilli. An aggregate of 60 bacterial strains was identified and characterized based on morphological, biochemical, and 16S rRNA gene sequencing from the submerged rice rhizosphere (*Amruta et al., 2016*). Microbes are positively correlated within water films for proliferation and active soil metabolics. The gram-positive bacteria's dominated the colonies of various PGPR isolated from chilli rhizosphere and efficiently resisted the noxious wilt disease of chili (*Yanti et al., 2017*). *Haque, Khan & Ahamad (2018)* in his investigations had characterized twelve strains of biocontrol isolates, including *Aspergillus niger, Trichoderma harzianum, T. viride, P. lilacinus, B. subtilis, P. fluorescens* and *P. putida* from the soil samples collected from the rice rhizosphere of Aligarh. *Houng, Padgham & Sikora (2009)*, in a parallel study of rice cropping areas in Vietnam, isolated *Fusarium* spp. and *Trichoderma* isolate from alluvial, acid sulphate and glyic acrisol soil of

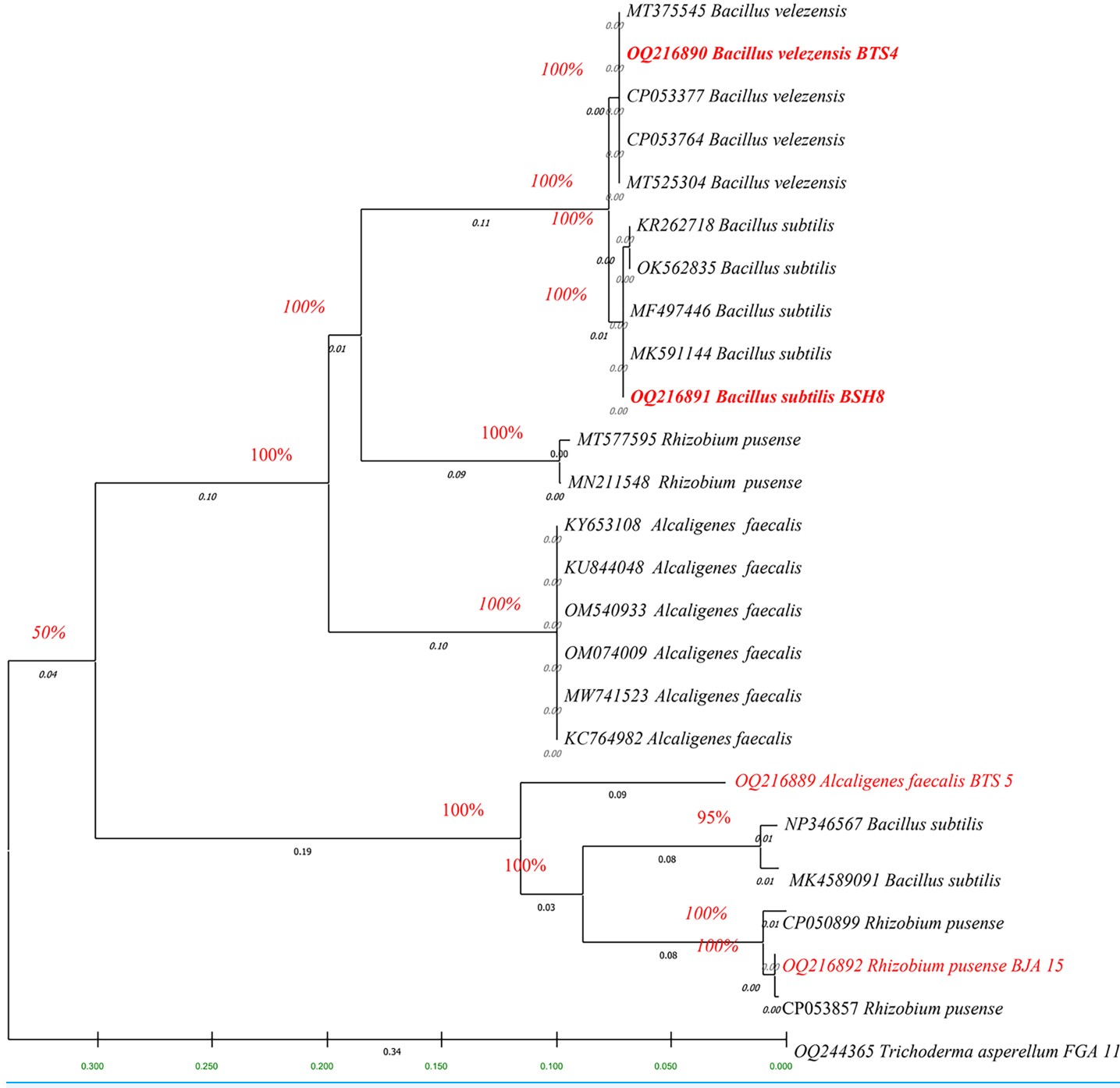

**Figure 2 Phylogenetic tree of potential bacterial strains with out-group and boot strap values.**

rice. The strains of *B. megatarium*, *B. subtilis*, *T. asperellum*, *Alcaligenes faecalis* and *B. velezensis* were isolated from medicinal, aromatic plants, cocoyam, cacao banana crop and oil seed rhizosphere (*Tondje et al., 2007*; *Torres et al., 2020*).

Efficient bacterial strains of fluorescent *Pseudomonads* habitat rhizospheric soils of chilli, field bean, green gram, brinjal, sunflower, red gram, groundnut, tomato, bermuda

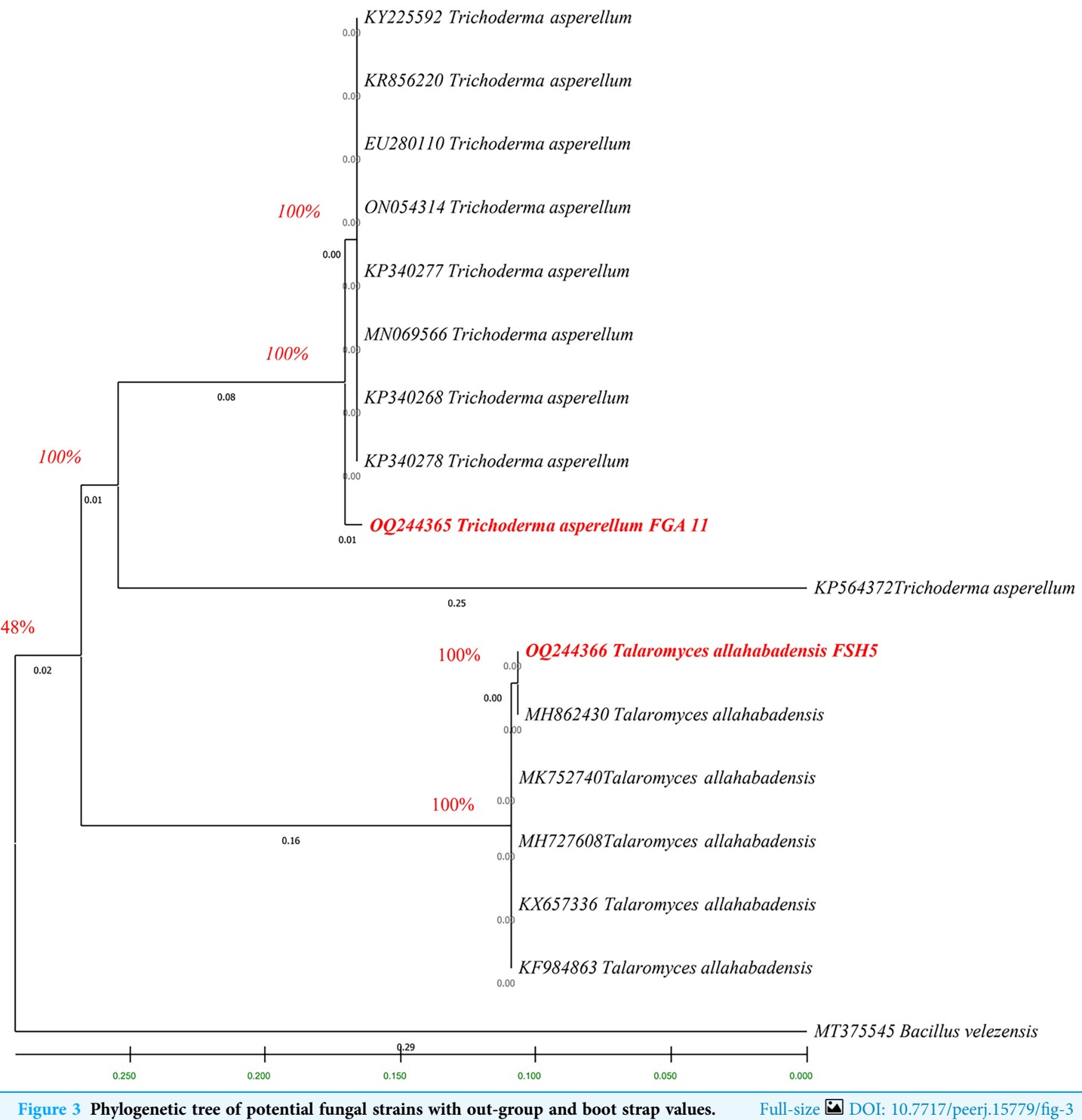

**Figure 3  Phylogenetic tree of potential fungal strains with out-group and boot strap values.**

grass, beans, sorghum, paddy and sesame (*Manjunatha et al., 2012*). Furthermore, bacterial strains were reported in the rhizosphere of soils with nematode outbreaks (*AbdelRazek & Yaseen, 2020*). The efficient bacterial and fungal microbial strains, later *to invitro* tests, through 16S RNAwere identified to be *B. subtilis* (OQ216891), *Bacillus*

*velezensis* (OQ216889), *Alcaligenes faecalis* (OQ216890), *Rhizobium pusense* (OQ216892), *Trichoderma asperellum* (OQ244366) and *T. allahabadensis* (OQ244365). Microbial-rich Indian soils nurture ubiquitous spp. of *Bacillus* and *Pseudomonads* and various reports support its efficient resistance to various biotic stresses, including economically salient nematodes (*Pandey et al., 2011*; *Khan, Mohiddin & Ahamad, 2018*; *Saikia et al., 2018*). The extended viability and forceful exertions of *Bacillus subtilis* against nematodes and their eggs facilitate its importance. *In vitro* results exhibited an average mortality of 80.79% against infecting juveniles of *M. graminicola* in our studies after 48 h. Numerous studies conclude that the infective juveniles of *Meloidogyne* spp. and plant parasitic nematodes including *R. similis*, *P. goodeyi*, and *H. multicintus* suffer the greatest mortality within 24 to 48 h of being inoculated with biocontrol agents under *in-vitro* conditions (*Ying et al., 2019*; *Hegazy et al., 2019*; *Kumar & Dara, 2021*). *Padgham & Sikora (2007)* isolated *B. megataium* from rice rhizosphere and tested it against *M. incognita* in controlled conditions and elucidated that the bio agent reduced the hatching of eggs of the tested nematode to 96% within 48 h. The crude cultural filtrate of *B. subtilis* releasing diffusible and volatile antibiotics, siderophores, cellulase, glucanase, protease, and chitinase substantiates the mortality of *M. graminicola. M. incognita, M. oryzae* (*Kumar et al., 2020*; *Amruta et al., 2016*). *Bacillus* spp. being an aggressive colonizer synthesizes metabolites that inhibit membrane protein and enzymes (*Schippers, Bakker & Bakker, 1987*) important against nematodes. The various antibiotics, proteolytic enzymes, high surfactin, and iturin activity supplemented the mortality of the juveniles *in vitro*. The strain's ability to deliver lytic enzymes such as chitinase, glucanase, and protease affects the nematode cuticle and survival (*Chen et al., 2015*; *Kavitha, Jonathan & Nakkeeran, 2012*). The phytonemtaodes are eliminated from the plant's rhizosphere by producing antimicrobial peptides, secreting lytic enzymes, vying for nutrients and space, and causing systemic resistance (*Kang, Radhakrishnan & Lee, 2015*).

Promising bacterial strain synonymized with *Bacillus amyloliquefacians* is now reclassified as *B. velezensis* (*Dunlap et al., 2016*; *Castro et al., 2020*), proliferates in a diverse ecology including oilseed plants (*Asaturova et al., 2021*) and black pepper (*Tran et al., 2022b*). The gram-positive, aerobic, endospore-forming beneficial bacterium was first reported from the mouth of river Velez in Malaga province of Spain (*Ruiz-García et al., 2005*). The salt-tolerant biocontrol strain *B. velezensis* diminished 71.29% of the juveniles of *M. graminicola* within 48 h. Antagonistic *Bacillus velezensis* BZR 277 activity was reported against the devastating root knot nematodes *M. incognita* in lab conditions (*Asaturova et al., 2021*). The resistance imparted against juveniles is sourced from the production of antibiotics and enzymes. *Bacillus velezensis* is a promising bio-agent for managing nematodes like *Meloidogyne* species because antibodies boost its chitinase activity (*Tran et al., 2022b*). Increased chitinase activity, or an antagonistic effect depending upon infestation pressure, facilitates the mortality of infective juveniles (*Lee & Kim, 2016*). *B. velezensis* synthesizes cyclic lipopeptides, such as surfactin, bacillomycin-D, fengycin, and bacillibactin, as well as polypeptides like macrolactin, bacillaene, and difficidin, that are harmful to nematodes (*Rabbee et al., 2019*).

*B. velezensis* produces lipopeptides identified through PCR amplification (*Azabou et al., 2020*) and a variety of extracellular metabolites, such as lipopeptides and biosurfactant molecules, have antimicrobial properties against many plant pathogens (*Ryu et al., 2004*). The presence of two chitinase genes chiA and chiB exhibited increased hydrolytic activity against colloidal chitin of nematode eggs (*Tran et al., 2022a*). Promising multifaceted biocontrol strain *A. faecalis* habitats in diverse and challenging conditions including oil spill soils (*Yadav et al., 2020*), *Mimosa calodendron* (*Felestrino et al., 2020*), *C. forskohlii* (*Mastan et al., 2020*) and okra crop (*Ray et al., 2016*). This represents the earliest instance of *A. faecalis* being observed in an Assamese rice crop rhizosphere; moreover, its adaptation to survive in extreme abiotic conditions makes it an interesting prospect for bio-control agents. RRKN juveniles population was alleviated and growth inhibited before exposure to *A. faecalis* in *in-vitro* conditions. Strains of *A.faecalis* reduced the hatching rates and improved mortality of the juveniles of *Pratylenchus* spp. and *Meloidogyne* spp (*Felestrino et al., 2020*). VOC's and antibiotic diffusates caused the mortality of *M. incognita* (*Lu et al., 2014*). Inhibtion of nematodes against the strain encompasses an array of antimicrobial substances, including identified Espprotein. This extracellular serine protease affected well over 85% of *M. incognita* juveniles' intestines (*Ju et al., 2016*). The BCA is reportedly adaptive to harsh environmental conditions and encodes HCN, siderophore, and phenolic compound synthesis, exhibiting mortality against nematodes (*Felestrino et al., 2020*). The dominance of chitin in nematode eggs and breakthrough for genes encoding chitinolytic, esterase activity, proteases and peptides (*Annamalai et al., 2011*), and production of dimethyl sulfate recognizes the strain as a future promising antinemic microorganism (*Xu et al., 2015*).

*Rhizobium* a genus dominant in pulses as PGPR was identified in the submerged rice rhizosphere. Avowed as an efficient strain for dissemination of potassium from mica *R. pusense* besides 21 phosphate-solubilizing bacteria was isolated from Egyptian agricultural soils (*Hauka, Afify & El-Sawah, 2017*). Indian ecology dominant food crops such as potato, pigeon pea, maize, banana, sugarcane, tobacco shares *R. pusense* (Phospate solubilizing rhizobacteria) rich microbiome that exhibits potassium solubilizing ability (*Meena et al., 2015*). The application of potassium (K) demonstrates that the K treatment can lower the occurrence of harmful nematode infections in rice and boost crop output (*Liu et al., 2022*). *Rhizobium pusense* exhibited 68.33% mortality in *in vitro* conditions. The dynamic impact of the cultural filtrate of *R. pusense* resists the hatching of nematode eggs and antibiotics impair juvenile proliferation (*Khan, Mohiddin & Ahamad, 2018*). The released nematicidal compounds resist *Meloidogyne javanica* at varying degrees (*Parveen et al., 2019*). Toxin identified as rhizobitoxine synthesized by *R. japonicum* aggregates the nematode pathogenesis (*Siddiqui, Baghel & Akhtar, 2007*).

Antagonizing microbe, *Trichoderma spp.* are becoming widely recognized as a means of restraining plant diseases (*Dukare et al., 2019*). *T. asperellum* isolated from chilli rhizosphere encompasses all ecological niches signifying itself as ubiquitous. Fungal isolates, including *T. asperellum* dwells in the rhizosphere of cocoyam, cacao and banana crop (*Tondje et al., 2007*), sugar beet rhizosphere Egypt (*Gueye et al., 2020*) and mango orchard, Mexico (*dos Santos Pereira et al., 2021*). Undistrubed ecosystem reserves

unexplored and potential microbes. The northeastern region harbors *Trichoderma sp.* is partially explored in pockets of Manipur (*Kamala & Devi, 2012*), the tea gardens of Assam (*Naglot et al., 2015*), and sorghum plants of Uttrakhand (*Manzar et al., 2021*). *Meloidogyne spp.* is effectively parasitized by *T asperellum* through inhibited egg hatching and direct larval mortality (*Kumari, Sharma & Baheti, 2020*). In bioassay conditions, MISC antibodies released from the fungal strain regulate improved juvenile death (*Sharon et al., 2009*). Arsenal of antagonistic activities against the nematode by *Trichoderma* species was reported to contribute to the biological defiance, including nutrient and space competition, antibiosis, mycoparasitism, and induced systemic resistance of plants (*Lombardi et al., 2018*). *Trichoderma* spp. excrete several lytic enzymes (glucanases, chitinases, proteases and lipases) to degrade cell wall components of other pathogens. The transcriptional activity of chi18-5 (chit42) and chi18-12 (chit33) of *Trichoderma* regulates chitinolytic enzyme systems during egg parasitism (*Szabó et al., 2012*). *Trichoderma* spp. produces nematicidal compounds such as trichodermin and trypsin-like protease, exhibiting larvicidal and ovicidal activities against the nematodes (*Yang et al., 2012*). The mortality *in-vitro* could be facilitated due to conidial attachment to the nematode body thus immobilizing the host, antifungal compounds (*Sharon et al., 2009*). The native strains considered safe for mammals promise faster growth rate (31–47 cm), finer conidial arrangement, and heat resistance at 30 °C as compared to *T. viride* with 11–33 cm growth radius and suitable growth at 11 °C (*Samuels, Lieckfeldt & Nirenberg, 1999*).

*Talaromyces allahabadensis* parasitised the nematode juveniles and restricted its movement leading to a mortality of 66.24%. Seventy-three fungal isolates collected from different regions, including *Talaromyces assiutensis* parasitized all the juveniles of *M. javanica* in an *in-vitro* predation test and killed 30–50% of the juveniles (*Hamza et al., 2017*). The exhibited potential mortality is due to novel potent nematicidal thermolides (*Guo et al., 2012*). The varying genus of economic phytonematodes, including *Pratylenchus zeae* falls prey to *Talaromyces* sp. (*Kisaakye, 2014*).

## CONCLUSION

The rhizospheric soils are rich in microorganisms and, in our study, 18 bacterial strains and 11 fungal strains were isolated from soil samples collected from the rhizosphere of crops like rice, okra, ash gourd, chili, beans, cucumber from diverse conditions and soil types in Upper Brahmaputra Valley region of Assam. The isolated native strains efficiently resisted the growth and population dynamics of infecting RRKN in bioassay conditions. Native biocontrol agents were hardly explored against nematode pests and the study will broaden the bioagent base against the devasting nemic pest. Combined treatment of the bacterial and fungal bio-control agents proves to be efficient against the rice root-knot nematode. Molecular insights into the activation of genes for the production of antibiotics, metabolities, enzymes and nematode trapping mechanisms are to be deepened. The multidimensional facets of promising identified microbes enhance the changes of successful establishment of the BCA's in the soil, which is of primary concern. *Alcaligenes faecalis* being salt tolerant, can be a viable option in the present soil deteriorating and challenging abiotic conditions. *T. asperellum* ability to increase rapidly in harsh conditions

improves its efficiency against abiotic stress. The VOC's and antibiotics released against *M. graminicola* are to be studied for further insight into the mechanism. Further, the native strains could be integrated into studies on nematode management in field conditions and could provide encouraging insight into integrated pest management with a synergistic impact on environmental sustainability.

## ABBREVIATIONS

| | |
|---|---|
| **RRKN** | Rice root knot nematode |
| **BGRIE** | Bringing Green Revolution to Eastern India |
| **ICR** | Institutional cum research |
| **UBVA** | Upper Brahmaputra Valley region of Assam |
| **rpm** | revolutions per minute |
| **DNA** | Deoxyribonucleic acid |
| **RNA** | Ribonucleic acid |
| **PCR** | Polymorphic chain reaction |
| **BLAST** | Basic Local Alignment Search Tool |
| **VOC's** | Volatile Organic Compound |
| **BCA's** | Biocontrol Agents |
| **PGPR** | Plant growth promoting rhizobium |
| **HCN** | Hydrogen cyanide |

### Funding
The authors received no funding for this work.

### Competing Interests
Ravinder Kumar is an Academic Editor for PeerJ.

### Author Contributions
- Rupak Jena conceived and designed the experiments, prepared figures and/or tables, authored or reviewed drafts of the article, and approved the final draft.
- Bhupendranath Choudhury conceived and designed the experiments, performed the experiments, authored or reviewed drafts of the article, and approved the final draft.
- Debanand Das conceived and designed the experiments, performed the experiments, analyzed the data, authored or reviewed drafts of the article, and approved the final draft.
- Bhabesh Bhagawati conceived and designed the experiments, performed the experiments, analyzed the data, authored or reviewed drafts of the article, and approved the final draft.
- Pradip Kumar Borah conceived and designed the experiments, performed the experiments, analyzed the data, authored or reviewed drafts of the article, and approved the final draft.

- Seenichamy Rathinam Prabhukartikeyan conceived and designed the experiments, performed the experiments, prepared figures and/or tables, authored or reviewed drafts of the article, and approved the final draft.
- Swoyam Singh conceived and designed the experiments, performed the experiments, authored or reviewed drafts of the article, and approved the final draft.
- Manaswini Mahapatra performed the experiments, authored or reviewed drafts of the article, and approved the final draft.
- Milan Kumar Lal analyzed the data, prepared figures and/or tables, authored or reviewed drafts of the article, and approved the final draft.
- Rahul Kumar Tiwari analyzed the data, prepared figures and/or tables, authored or reviewed drafts of the article, and approved the final draft.
- Ravinder Kumar analyzed the data, prepared figures and/or tables, authored or reviewed drafts of the article, and approved the final draft.

## DNA Deposition

The following information was supplied regarding the deposition of DNA sequences:

The data is available at GenBank: OQ216889 (BTS4 *Bacillus velezensis*), OQ216890 (BTS5 *Alcaligenes faecalis*), OQ216891 (BSH8 *Bacillus subtilis*), OQ216892 (BJA15 *Rhizobum pusense*), OQ244365 (FSH5 *Talaromyces allahabadensis*), OQ244366 (FJB11 *Trichoderma asperellum*).

## Data Availability

The raw data are available in the Supplemental Files.

## Supplemental Information

Supplemental information for this article can be found online at http://dx.doi.org/10.7717/peerj.15779#supplemental-information.

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
