# Peer review of "Diversity of bioprotective microbial organisms in Upper Region of Assam and its efficacy against Meloidogyne graminicola"

_PeerJ, doi:10.7717/peerj.15779_

## Round 0.1 · original submission · Major Revisions

Dear Author,
Kindly go through the suggestion/queries raised by both reviewers. Incorporate them and revise the manuscript.

·

Basic reporting

References are not according to the journal's format. Majority of work needed to be done on references.
Discussion is not appropriate and need special attention to rewrite it according to the results.
Few photos related to the work will be good and improve the quality of this paper, if available.
"et al" should be italicized everywhere.

Experimental design

Experimental design is upto the mark.

Validity of the findings

The findings are validating according to the present scenario.

Reviewer 2 ·

Basic reporting

Overall, the manuscript is interesting and well written

Experimental design

There are number of shortcomings in the methodology section.
For example,
Why do you centrifuge after passing through bacterial filter?
How did the authors choose for 28 ˚C for 48h for growing bacteria?
Will the section 2.6 be reproducible, since the production of metabolites might vary and just diluting might not be a right way of quantification?
I suggest the authors to recheck the methodology section.

Validity of the findings

The data in the manuscript is just the screening of microbes which is preliminary study. The authors should work on the mode of action or mechanism.

Additional comments

What is the rationale for collecting soil from plant growing areas other than paddy?
The mechanism of inhibition should be studied
Discussion needs to be improved based on the results obtained and keep it to the point.
Conclusion should highlight the novelty of work.
Abstract should be improved
Please expand RRKN and IPM in abstract

---

## Round 0.2 · accepted · Accept

Dear Authors,
It is pleasant to me to accept the manuscript "Diversity of bioprotective microbial organisms in Upper Region of Assam and its efficacy against Meloidogyne graminicola" based on reviewer's report. It is quite impressive.
Congratulations.
With Regards
Ravindra

·

Basic reporting

Author has considered all the suggestions and incorporated it in the manuscript. Article structure, figures and tables are nice and needs no further revision. Result is well written and easy to understand.

Experimental design

No improvement is needed.

Validity of the findings

All the experiments are validated.